# Extended Energy-Expenditure Model in Soccer: Evaluating Player Performance in the Context of the Game

**DOI:** 10.3390/s22249842

**Published:** 2022-12-14

**Authors:** Arian Skoki, Alessio Rossi, Paolo Cintia, Luca Pappalardo, Ivan Štajduhar

**Affiliations:** 1Department of Computer Engineering, Faculty of Engineering, University of Rijeka, Vukovarska 58, 51000 Rijeka, Croatia; 2Department of Computer Science, University of Pisa, Largo B. Pontecorvo 3, 56127 Pisa, Italy; 3Institute of Information Science and Technologies (ISTI), National Research Council of Italy (CNR), Giuseppe Moruzzi 1, 56124 Pisa, Italy; 4Center for Artificial Intelligence and Cybersecurity, University of Rijeka, R. Matejcic 2, 51000 Rijeka, Croatia

**Keywords:** game intensity, clustering, machine learning, fatigue, fitness tracking

## Abstract

Every soccer game influences each player’s performance differently. Many studies have tried to explain the influence of different parameters on the game; however, none went deeper into the core and examined it minute-by-minute. The goal of this study is to use data derived from GPS wearable devices to present a new framework for performance analysis. A player’s energy expenditure is analyzed using data analytics and K-means clustering of low-, middle-, and high-intensity periods distributed in 1 min segments. Our framework exhibits a higher explanatory power compared to usual game metrics (e.g., high-speed running and sprinting), explaining 45.91% of the coefficient of variation vs. 21.32% for high-, 30.66% vs. 16.82% for middle-, and 24.41% vs. 19.12% for low-intensity periods. The proposed methods enable deeper game analysis, which can help strength and conditioning coaches and managers in gaining better insights into the players’ responses to various game situations.

## 1. Introduction

The intensity of a soccer game is dependent on a wide range of factors: quality of the opposition, period of the season, weather, team form, game status, etc. There is a lot of work dealing with the influence of various parameters on said factors—such as location, opponent quality, and game outcome—by using the most important metrics derived from wearable sensors (e.g., total distance, accelerations, and decelerations). However, one of the most dominant factors in soccer is the quality of the opposition. Higher-quality opponents usually require higher physical demands during the game, which, in turn, results in increased values of the total distance (TD), maximal speed, average speed, frequency of high-intensity actions (HIAs) [1], and events related to changes in velocity (accelerations and decelerations) [2]. This is not always true, because it heavily depends on the context and the play styles of each team. According to Garcıa-Unanue et al. [3], away games accumulate significantly more TD (+230.65 m, 95% IC: 21.94 to 438.19, ES: 0.46, *p* = 0.031), but there were no differences found that depended on the opponent quality. However, an analysis between the first and second half revealed a significant reduction in TD covered by the players against lower-level teams (−290.42 m, 95% CI: −557.82 to −23.01, ES: 0.72, *p* = 0.033) and medium-level teams (−374.56 m, 95% CI: −549.21 to −199.70, ES: 0.71, *p* < 0.001). Congested periods can also have a great impact on player performance and reduce the number of HIAs that a player can sustain [4]. Differences in intensities are present across playing positions [5]. Therefore, central midfielders (CM) and wide midfielders (WM) cover greater TD and have higher average speed, whereas WMs and fullbacks (FB) cover higher distances at high-speed running (HSR) and sprinting [6]. Center backs (CB) and forwards (FW) usually cover the smallest distances [7], but FWs cover more HSR distance than CMs [6]. Concerning the game outcome, recent work reported higher workloads in won games compared to losses and draws, higher TD and HSR distance in the second half of defeats, and lower average speed in wins compared to draws and losses [8]. When looking at score change influence on the game demands, there have been reports that CMs, WMs, and FWs have increased TD while winning, but CBs and WBs have higher TD and HSR while losing [9,10]. Interpretation of scoreline influence needs to be done carefully, because there is a need for a clear scoreline definition for which to consider scoreline effects [11].

Several studies [3,6,7,11,12] have described various context situations and their effects on the wearable-derived metrics. Nevertheless, there were no attempts to increase the level of data sampling and examine physical demand change in a game on a minute-by-minute basis. Another problem is the usage of different GPS providers, which have various thresholds for determining sprint distance, HSR distance, number of sprints, number of detected acceleration and deceleration phases, etc.; hence, the concept cannot be easily transferred. Nowadays, most providers allow users to set their own preferred thresholds. However, this invokes other problems, such as many different thresholds used for the same parameters (no consensus). In addition, accumulated values (which are the most-used ones) provide an overall image, but often that is not particularly useful as it represents an average of the game. A team can play an average-intensity game overall but actually perform a very intensive first half and a less-intensive second half of the game [8]. This information is very important, along with individual analysis of a particular player’s effort within the game. Intensity inspection within the context of scored and conceded goals can provide information about which player could be a better substitute depending on the game scoreline. Not every game is the same—each one provides unique demands [13]. Therefore, the training load in the preceding week should be adjusted according to expenditure in the previous game, and player fitness needs to be tracked regularly to evaluate season meso- and macro- cycles [14]. For this purpose, better methods for game analysis are needed than the ones that are currently used.

Hence, the aim of this study is to minimize the effect of various GPS provider thresholds by taking an energy expenditure approach. The study shows how to develop a framework of data analytics for evaluating workload intensity as the game goes on. In particular, this enables a detailed examination of the workload throughout the game and provides a baseline for better understanding of the game demands depending on the context. Moreover, methods proposed in this study can be used for objective tracking of the players through meso- and macro-cycles of the season.

## 2. Materials and Methods

### 2.1. Study Design

The data were acquired during official and preseason games of a professional soccer club. The process of data collection was executed by using GPS wearable sensors, GPexe pro2 (Exelio Srl, Udine, Italy), with a sampling rate of 18 Hz. According to position, the players were divided into five categories: center back (CB), wing back (WB), midfielder (MF), wide forward (WF), and forward (FW). The sensor provider enabled the usage of two types of data: (1) GM-5MIN (GPS metrics of expenditure through 5 min intervals) and (2) metabolic power events (MPEs), (HIAs, which occur throughout the whole course of a game).

### 2.2. Subjects

In total, there were 38 male soccer players (age 25±3 years; height 1.81±0.06 m; weight 76±5 kg) that played at least one game during the acquisition period. The goalkeeper (GK) position was not recorded and therefore it was not used for analysis. All the players that were wearing the sensor during the games were included in the dataset. This applies to the substitute players, too. Playing positions counted 11 CBs, 7 WBs, 8 MFs, 3 WFs, and 9 FWs. The club allowed the research team to access players’ data, and informed consent was provided.

### 2.3. Data Acquisition

The data collection process started in January 2021 and ended in March 2022, which rounded a period of two half-seasons (including preseasons) and included 80 games. The game data were acquired using GPexe pro2 devices that collected all of the standard parameters for analysis [15] with the addition of metabolic expenditure features [16]. Furthermore, GPS metrics derived from the whole game (90 min data) were used as a baseline for explaining and validating the descriptive power of the proposed clustering methods. In the rest of the paper, we refer to this data type as GM-GAME.

### 2.4. Procedures and Variables

Before each game, players would put on the GPexe pro2 device, which was located in a wearable vest. The sensor notified the wearer with a red blinking light when there was a need for re-calibration. This was done very easily by spinning the device for a couple of seconds. After the match, collected data were downloaded from the device and uploaded to the manufacturer’s platform using the GPexe bridge application (version 8.3.6). The online web application (version 7.4.46) computed all the metrics in a 5 min sampling window. In order to minimize problems due to different thresholds and varying results, only energy-based metrics and TD were used in the analysis. This included: total time played (min), distance (m), average metabolic power (W/kg), energy (J/kg), anaerobic energy (J/kg), MPE count, MPE average recovery time (s), MPE average recovery power (W/kg), walk distance (m), running distance (m), walk energy (J/kg), and running energy (J/kg).

Average metabolic power was calculated by multiplying speed and energy cost (the description of parameter calculation was taken from the GPexe dictionary for athletic performance monitoring, which is an internal document accessible only to their users), which has been thoroughly described by di Prampero et al. [16]. The energy variable is an estimation of both the energy required to cover a given distance at a constant speed and the energy needed to perform speed variations. The same is true for calculating anaerobic energy, with a difference of taking into account the player’s maximal $VO_{2}$ as a measurement threshold [17]. The MPEs are defined as phases during the exercise (or a game) based on a difference between the estimated metabolic power and oxygen consumption (the description of parameter calculation was taken from the GPexe dictionary for athletic performance monitoring, which is an internal document accessible only to their users). Since the maximal $VO_{2}$ of each athlete can be directly or indirectly assessed, this value allows individual analysis and overcomes limitations of other models that are based on specific speed or acceleration thresholds. The MPE recovery (power and time) is detected by the power decrease that happens in order to repay previously contracted oxygen debt (the description of parameter calculation was taken from the GPexe dictionary for athletic performance monitoring, which is an internal document accessible only to their users). The features regarding walking and running are not defined by a fixed speed threshold but depend rather on different combinations of speed and acceleration [16].

All the presented features are shown in Table 1. Many of them were related to metabolic power for the reason of avoiding difficulties involving speed, acceleration, and deceleration threshold values [17]. The accounted features were aggregated and derived from GM-5MIN and MPE data, which are described thoroughly in the following sections.

#### 2.4.1. MPE Data

A special feature of GPexe wearables is the focus on metabolic expenditure using MPE. The main difference and benefit of the approach using MPE is that it does not take into account only acceleration and deceleration for detecting HIA [16], but instead it is focused on energy expenditure; thus, the problem of setting the threshold is avoided. An issue regarding the acceleration approach lies in setting up a threshold for detecting these events, which heavily influences the resulting values. Moreover, there exists a lack of agreement and information regarding the choice of methods for acceleration filtering [18]. Instead, the alternative approach proposed by di Prampero et al. in 2005 is used [19], which sets the standard for calculating MPE. This approach assumes that accelerated running on flat terrain is equivalent to constant running uphill at a constant speed at a certain angle. Power events happen often in the game, and they are the most important factor in energy expenditure. In the full game, there are more than 100 MPEs per player, which differ in duration and power. An example of the data for one player and a single game can be seen in Figure 1. The MPE dataset contained information about the start and end timestamp of an event, duration in seconds, maximal speed, and average power spent. The energy expenditure of an MPE was calculated by multiplying the average power and duration of the event, which is shown in Equation (Equation 1).
(1)$$E_{MPE}=P_{avg}*t$$

This could be done because most of the events are short in duration, up to 20 s, with a median value of 5.8, a mean value of 6.5, and a standard deviation of 3.35 s. The distribution of event duration can be seen in Figure 2. The described data provide information about the peak energy expenditure. However, this is not enough for a complete understanding of expenditure within a game because there are no data about the period in which a player was recovering (see Figure 3). Data regarding what is happening in the recovery period are lacking; therefore, if one would use only MPE data, then only the information about the HIA would be considered. The reality is that recovery can be passive or active: a player can stand, walk, jog or run. This information is crucial for understanding in-game player recovery.

#### 2.4.2. GM-5MIN Data

MPE data occurrence is discrete across the whole game. That means that these events (short in duration) are always separated by periods of recovery. The energy that is spent during MPEs is equivalent to 30% of the total energy consumption (the description of parameter calculation was taken from the GPexe dictionary for athletic performance monitoring, which is an internal document accessible only to their users), which can be seen in Figure 4. The remaining 70% of energy expenditure is ignored by this type of data. The lack of information about a player’s expenditure during the recovery periods is addressed with the introduction of GM-5MIN data, which contains an average of values in a given 5 min period. All the events within the 5 min interval were hence taken into consideration, including walking, jogging, running, sprinting, TD, energy, etc. This enabled the acquisition of the full 100% of energy expenditure within the observed period. An example of the GM-5MIN and MPE energy expenditure for one player and a game can be seen in Figure 4. Ideally, one would like to have these values in the 1 min interval and thus have a detailed image of expenditure. Unfortunately, due to the intrinsic limits of the GPS tracking device, the processing time for shorter intervals exponentially increases with regard to the duration of the interval and could not be extracted in a reasonable time. This is the reason for settling on a 5 min period: it is short enough to capture game details but also long enough for calculating accumulation metrics such as average metabolic power, average recovery power, etc.

### 2.5. Data Preprocessing

To fully describe the intensity of a game, the MPE and GM-5MIN parameters needed to be combined together. This was done by iterating through each game for all the players. As the exact times and durations of both MPEs and GM-5MIN were known, they could be combined. The process consisted of merging three main data sources, which included: (1) GM-5MIN features in the preceding 5 min period (GM-5MIN-PRIOR), (2) MPE features in the preceding 3 and 5 min periods (MPE-PRIOR), and (3) MPE features in the observed minute (MPE-CURRENT). The entire processing workflow is shown in Figure 5.

The resulting dataset counted 25 parameters, which are described in Table 1. The process to create GM-5MIN-PRIOR consisted of using the GM-5MIN data of a 5 min period that preceded the current processing minute. The aim of this was to measure the overall expenditure just before the observed minute. The second step was adding MPE-PRIOR features, which gave information about the peak intensity in the period preceding the observed minute. Finally, information about the observed minute was added by the introduction of MPE-CURRENT features.

**Table 1 sensors-22-09842-t001:** A list and description of features used for clustering. Features are divided into 3 categories depending on the data source: GM-5MIN-PRIOR, MPE-PRIOR, and MPE-CURRENT.

	Feature Name	Description
GM-5MIN-PRIOR	Distance (m)	Distance covered in the last 5 min.
	MPE count	Number of MPEs in the last 5 min
	Anaerobic energy (J/kg)	Anaerobic energy spent in the last 5 min
	Average metabolic power (W/kg)	Average metabolic power spent in the last 5 min
	Average MPE time (s)	Average MPE duration in the last 5 min
	Average MPE recovery time (s)	Average recovery time in the last 5 min
	Average MPE recovery power (W/kg)	Average recovery power in the last 5 min
	Walk energy (J/kg)	Energy spent walking in the last 5 min
	Running energy (J/kg)	Energy spent running in the last 5 min
	General energy (J/kg)	Energy spent on all activities in the last 5 min
	Total number of MPEs	Number of MPEs up to that moment in the game
	Total energy spent (J/kg)	Energy spent up to that moment in the game
MPE-PRIOR	MPE energy (3 min)	Energy spent on MPEs in the last 3 min
	MPE energy (5 min)	Energy spent on MPEs in the last 5 min
	MPE count (3 min)	Number of MPEs in the last 3 min
	MPE count (5 min)	Number of MPEs in the last 5 min
	Recovery time (s) (3 min)	Recovery time (s) in the last 3 min
	Recovery time (s) (5 min)	Recovery time (s) in the last 5 min
	Average recovery time (s) (3 min)	Average recovery time (s) in the last 3 min
	Average recovery time (s) (5 min)	Average recovery time (s) in the last 5 min
	Total recovery time (s)	Recovery time up to that moment in the game
MPE-CURRENT	MPE energy spent (J/kg)	Energy spent on MPEs in the observed minute
	Event count	Number of MPEs in the observed minute
	Average recovery time (s)	Average recovery time in the observed minute
	Recovery time (s)	Recovery time in the observed minute

On top of described variables in Section 2.4, additional ones were derived from these data and incorporated into the feature set. In order to keep information about the duration of the game and the influence of fatigue, cumulative (total) features were created. This group of features comprised: the total number of MPEs, total energy spent (J/kg), and total recovery time (s). Information about MPE average recovery time was only available for a 5 min period preceding the observed minute. To get information about the average recovery time in the 3 min preceding and the observed minute, a new metric was derived from MPE data. The metric represents absolute recovery time in seconds (within the observed period) divided by the number of events in that period. An example is shown in Figure 3, where there are 4 MPEs and 5 recovery periods, which gives 18 s of work and 42 s of recovery. An average recovery time is thus equal to Equation (Equation 2), which gives an average of 8.4 s for the observed example.
(2)$$t_{avg\_recovery}=\frac{t_{recovery}}{N_{MPE}+1}$$

The MPE features were used to explain the work done in the current processing minute and also the last 3 and 5 min periods. Combined with GM-5MIN, the dataset had information about (1) overall energy consumption and recovery time in the preceding 5 min, (2) MPE energy consumption and recovery time in the preceding period of 3 and 5 min, and (3) MPE energy consumption and recovery time in the current minute. After all the players and games were processed, extreme values for every feature were limited to fit the ceiling value in order to denoise the data. This value was determined by calculating the threshold at which 99.5% of the data would fit in that range. All the values that were higher than the ceiling threshold were limited to that threshold.

### 2.6. Clustering Analysis

To account for the scarcity of data instances (limited number of games and players) and to suppress overfitting, the number of features was reduced to successfully apply the clustering algorithm. We used principal component analysis (PCA) for dimensionality reduction. Before that, each feature was normalized by applying a min–max normalization. The next step was to divide the resulting data points (after min–max normalization) into different categories. As this was a new approach and the labels were unknown beforehand, an unsupervised clustering algorithm had to be used. For this purpose, K-means was selected to evaluate how many intensity zones were present in the dataset. Before fitting centroids on the data, halftime and the start of the game needed to be excluded from training. By analyzing the data, a threshold of 200 m was set as the minimum distance; thus, all the instances that were lower than that value were ignored. Inspection showed that these outlier distance values were part of the halftime recovery and the first 5 min of the game (lacking information about the energy expenditure in the 5 min that preceded). The chosen threshold of 200 m enabled the exclusion of all the outlier values. Failure to do so would have heavily affected the results for the low-intensity zone, which is described later in Section 3.2. The K-means algorithm was tested for k∈[2,3,4,⋯,15] clusters using the within-cluster sum of squares (WCSS). The best number of clusters was defined through the elbow method on WCSS values. Clustering analysis is a prerequisite for better understanding the physical demands of the game and for future research about the influence of various game context variables on players’ physical behavior.

### 2.7. Clustering Application

After the dataset had been created (see Figure 5), we clustered intensity profiles (groups) that enabled assessment of the effort performed by the players as the game goes on. Moreover, a Markov chain analysis was conducted in order to estimate players’ capabilities of intensity shifts during a game. This part of the paper may be more interesting for specific readers; therefore, a detailed description of the methods used and the newly created MFit index can be found in the Appendix A. Appendix A explains the process of creating an MFit index, while Appendix A shows the results of such analysis by tracking the players’ fitness through meso- and macro- cycles of the season.

## 3. Results

In this section, we explain how the proposed clustering method can be used for detailed minute-by-minute game analysis that can be performed both individually or on a team basis. It also provides an example of game load comparison in Section 3.3, with the premise that each game has unique physical demands.

### 3.1. Clustering Analysis Results

PCA analysis showed that the optimal number of components for dimensionality reduction is seven, preserving 92.8% of data variance (see Figure 6). Table 2 presents the most important features of every component, with the minimal importance of a single feature being 0.3 (explaining 30% of the data). Based on these seven principal components, the elbow of the score was obtained by analyzing a different number of classes by the K-means algorithm. We determined that the optimal number of clusters was three (Figure 6), corresponding to low, middle, and high intensities in the game. The most important features and their distributions across clusters are shown in Table 3.

The K-means algorithm assigns each particular minute in the game to one of the three possible groups. It is expected that higher intensity causes more energy consumption and an increased number of MPEs but less time spent in recovery. To distinguish between low, middle, and high intensity, the groups needed to be examined in more detail. For this purpose, each feature distribution across classes was inspected by calculating the mean and standard deviation. As described in Section 2.5, clustering features were divided into three groups: (1) MPE-CURRENT, (2) MPE-PRIOR, and (3) GM-5MIN-PRIOR. The first one gives information about the current intensity, while the latter two explain what actually caused the current state. It can be seen that the low-intensity group does not contain any MPEs, but at the same time, it has the highest energy consumption in the last 5 min (GM-5MIN features—Energy (J)). The low-intensity group is a result of higher energy consumption in the preceding period and can be characterized as an active recovery period for the player.

### 3.2. In-Depth Game Visualization

Every game is unique to each player. Therefore, a visual representation of intensity zones throughout the game for a single player in a particular game is shown in Figure 7. It can be clearly seen that a player needs to make a stop after a certain number of high-intensity actions. The period following (middle and low intensity) could be a recovery period; however, the reader should note that it is dependent on the game context (e.g., penalty, set-piece, video-assisted referee decision). Scoring minutes are shown in order to better visualize the potential effect of scored and conceded goals on the game tempo. The distribution of the intensity zones across the span of the game is shown in Figure 8. It can be clearly seen that the high-intensity periods decrease as the game goes toward the end. At the same time, middle intensity periods start high and then slowly increase from the 10th min. Low intensity periods stay stable and slightly increase towards the end. This is mostly due to the effect of fatigue. The same inspection can be made for the whole team, but this requires additional analysis. Average minute-by-minute intensity can be aggregated by grouping playing positions and showing their average minute-by-minute intensity values for each position or by taking an average intensity for each minute using all the players in the team. It should be noted that this kind of analysis is inferior to individual inspection, and a lot of information can be lost in the aggregation.

### 3.3. Evaluation through Game Load

To show the effect and benefits of the proposed clustering approach, the results are compared with the GM-GAME data (each GPS feature separately) that is regularly used for athlete load monitoring. The premise that each game is unique was tested by looking at the distribution of high-, middle-, and low-intensity minutes for the whole team. The clustering algorithm provides information about the three groups (i.e., high, middle, and low), and the same thing needs to be done for GM-GAME data to enable comparison. Currently, by using GM-GAME data, there exists no consensus on how one could measure intensity using particular parameters. In order to compare the approach presented in this study with GM-GAME data, we need to draw parallels between the intensity clusters and a single GM-GAME feature. Therefore, the assumption was that GM-GAME data equivalents for the high-intensity group would be the distance covered above 20 km/h; for low intensity, it would be equal to walking distance (m); and for middle intensity, it would be equal to running distance (<20 km/h, walking excluded).

Table 4 provides the percentage of intensity groups in both the clustering algorithm and GM-GAME features. To compare the variability of each group, the coefficient of variation (CV) was calculated for each column. The results (row CV) show the variance in % for the particular column. The aim of this was to assess between-game variability and the potential for describing each game’s unique demands. Clearly, the clustering approach has superior CV results with the additional capability of detailed in-game inspection. Further, the distribution of values is more natural–mostly middle- and high-intensity clusters and a smaller numer of low-intensity clusters, which is a reasonably competitive game-demand distribution. On the other hand, GM-GAME puts focus on middle- and low-, with very little influence due to high-intensity. According to this, soccer players are not giving their maximal effort very often in competitive games, which is hard to believe. This confirms that tracking the intensity of the game is not a trivial problem, and that it is very hard to explain game intensity by using a single GPS parameter. Hence, based on these results and using only GM-GAME, we can speculate that there is no distinction between scoreline change, quality of the opponent, or some other context of the game. This shows the limitation of GM-GAME data and a need for better methods for intensity evaluation. A comparison of the MFit index approach (mentioned in Section 2.7) and GM-GAME parameters can be seen in Appendix A.

## 4. Discussion

This study provides a deep dive into game intensity in soccer by using data acquired from wearable sensors. Every game is unique and, therefore, should be treated independently. In the literature, a lot of work has analyzed the intensity concerning the scoreline [9,11,12,20]. This was done by comparing relevant metrics (e.g., HSR, distance, and sprint distance) depending on the score and also the quality of the opponent. However, all of the research was only focused on comparing the GM-GAME data from wearable sensors, which give overall information about the game. There have been no attempts to understand how players perform on a minute-to-minute basis and how fatigue influences the final minutes of the game.

To go deep into the game intensity, the proposed analytical approach takes into consideration two data sources (GM-5MIN and MPE data) as the game goes by. Surely, the possibility of including 1 min periods instead of 5 min periods would yield a better representation of the game. On the other hand, by using smaller periods of the game, accumulation error would go up, and this amount would vary between the providers [21]. Moreover, processing smaller intervals would be very long or even impossible with state-of-the-art tracking devices. To account for this, the framework of data analytics provided in this study enabled a detailed inspection of the game. In particular, the intensity of the game could be grouped into three main zones, i.e., low-, middle-, and high-intensity. As shown in Table 3, in low-intensity actions, players do not engage in any activity (recovery period), but this was a result of the preceding 5 min intense period. Differently, the middle- and high-intensity groups show medium and strong MPE activity, respectively, and shorter periods of recovery. However, the middle-intensity group shows low-intensity actions 5 min prior, while the high-intensity group is preceded by strong GM-5MIN expenditure. These groups enable us to analyze player status as the game goes on. As a matter of fact, Figure 7 and Figure 8 show intensity groups of a single player for the entire game. The figures suggest that there are several high-intensity periods in the starting part of the game, which recedes as the game goes by, indicating the fatigue status of the player or a change in the game intensity. Moreover, based on Figure 7, the game score seems to have no correlation with the intensity period. However, more research is needed to assess the effect of different intensity patterns on game scoring status, the tactical approach of a team, the roles of the players, and other game events.

The proposed framework provides a basis for additional analysis of the game context. This includes the changes in intensity cluster distribution depending on: the team’s possession style, the quality of the opponent, or the change of scoreline. By using more seasons and acquiring more data about the substitute players, a better analysis of their energy expenditure can be made. Currently, there are not enough substitute players with similar playing time and the same playing position. Further, with the addition of multiple teams with different playing styles, the model can be adapted according to e.g., possession or counter-attacking football. This, however, needs further investigation and is out of the scope of this study.

The applicability of the model was shown using in-game visualization of the player’s low-, middle-, and high- intensity clusters. The ability to provide better information about the game intensity compared to GM-GAME parameters was shown by calculating the CV of each cluster. The results proved that tracking the game intensity is not a trivial task, i.e., it cannot be observed from the aggregation level. However, the approach proposed in this study gives promising results because it takes into account minute-by-minute changes in energy expenditure. However, the reader should note that all of these analyses can be affected by the game characteristics, which can influence intensity and, consequently, physical demands.

The main limitation of this study is that only one team was used in the analysis. Therefore, we should inspect this on more teams to provide information about the accuracy and validity of this approach. However, in this study, we present a new approach for analyzing real-time intensity during a game that could be applied to each team. Hence, every team could create its own personalized model in accordance with individual players’ characteristics by applying our analytical approach.

## 5. Conclusions

This study is focused on diving deeper into players’ energy expenditure throughout the full length of a soccer game. It provides procedures for processing and clustering the data for the purpose of inspecting the physical game performance of the players on a minute-by-minute basis. This approach can help practitioners to better understand the specific context of the game (i.e., when the team decreases physical performance) but also provide ground for further inspection of fatigue, match context, and how the energy of the players is spent through the course of a game.

## Figures and Tables

**Figure 1 sensors-22-09842-f001:**
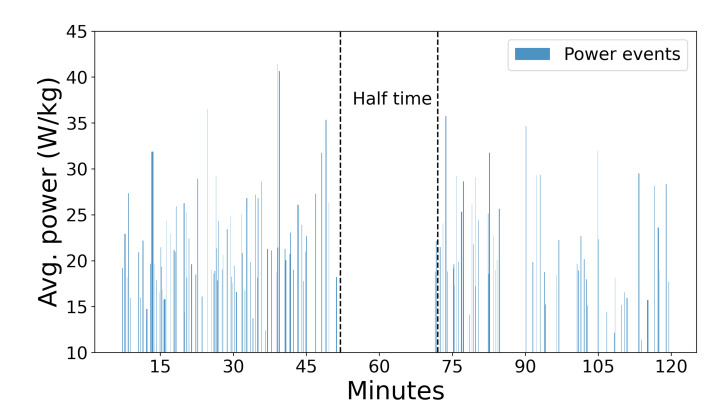
MPE in-game distribution for a single player.

**Figure 2 sensors-22-09842-f002:**
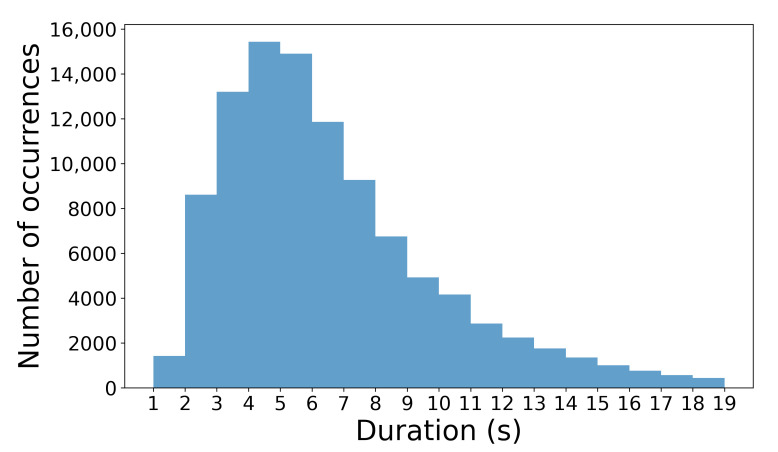
MPE duration histogram. The distribution median value is 5.8, the mean is 6.5, and the standard deviation is 3.35.

**Figure 3 sensors-22-09842-f003:**
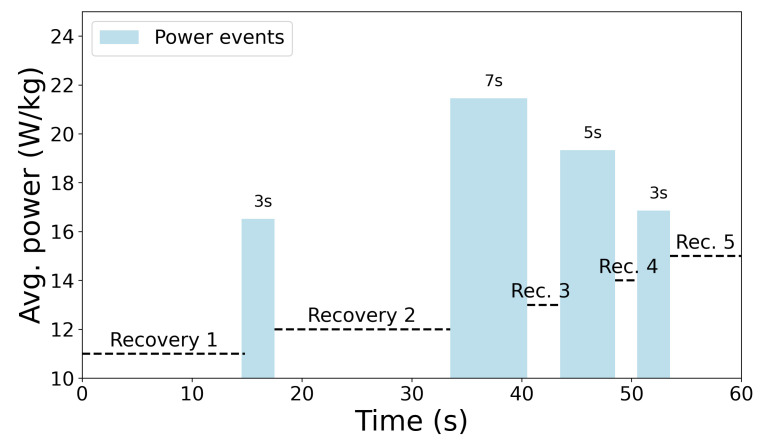
Recovery time within the power events (blue) in a 60 s time frame.

**Figure 4 sensors-22-09842-f004:**
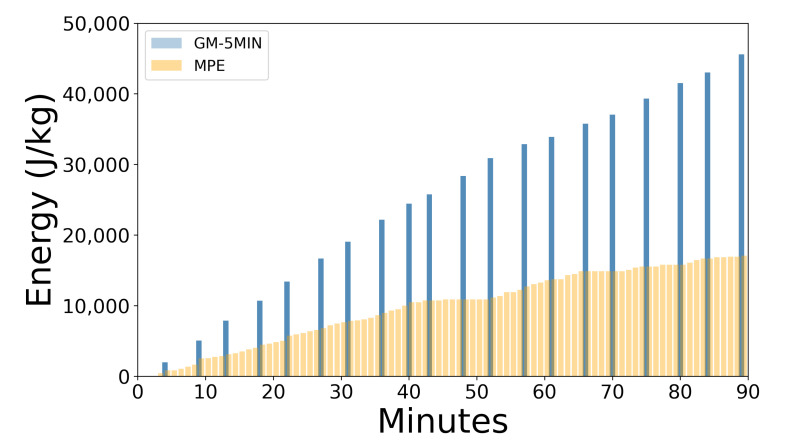
GM-5MIN vs. MPE expenditure across 90 min. Blue bar charts represent GM-5MIN across the 90 min of the game. Orange bars represent the energy expenditure of MPEs.

**Figure 5 sensors-22-09842-f005:**
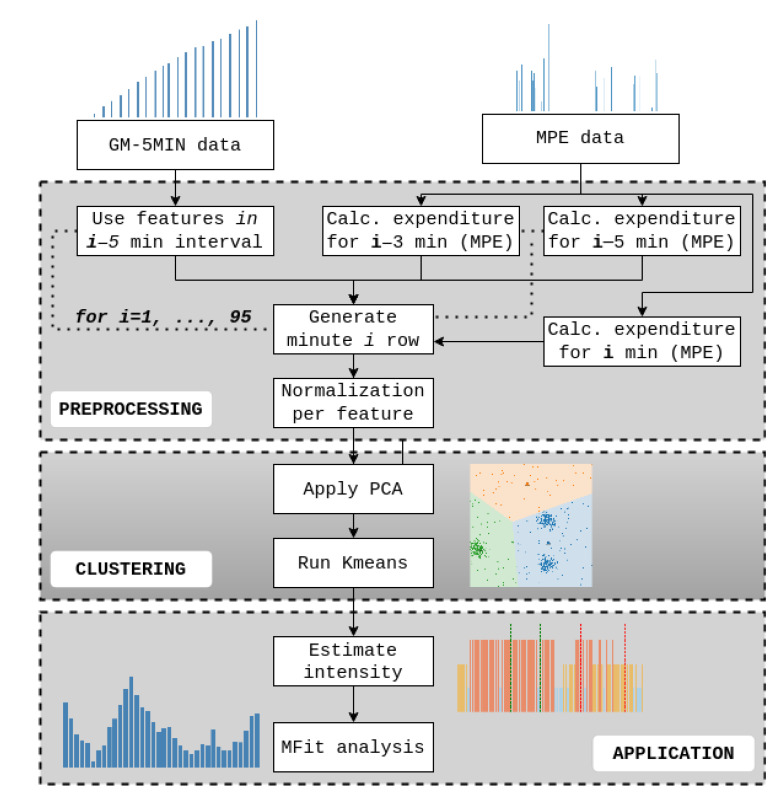
Preprocessing and clustering flowchart is divided into 3 parts. The first step shows how GM-5MIN-PRIOR, MPE-PRIOR, and MPE-CURRENT are combined. Next, dimensionality reduction using PCA is performed to prepare the data for K-means clustering. The final step consists of using the clustering algorithm to obtain low-, middle-, and high-intensity events throughout the game and to perform MFit analysis on the players.

**Figure 6 sensors-22-09842-f006:**
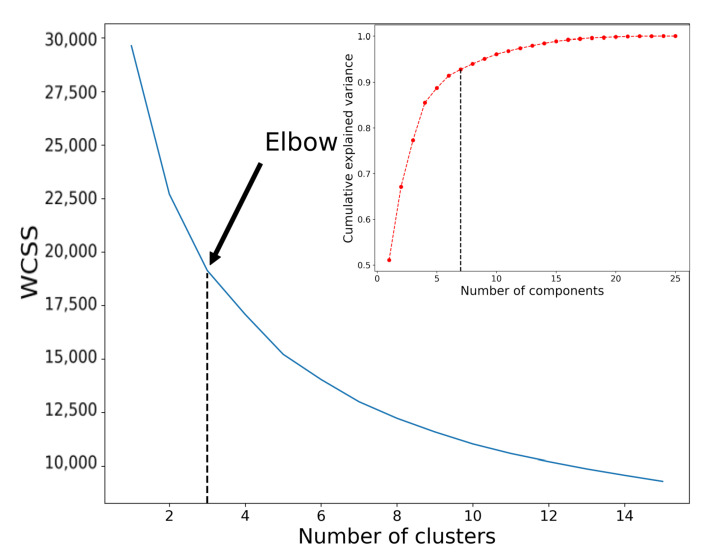
Analysis of K-means clustering with WCSS method and PCA component inspection. The optimal number of clusters is chosen using an elbow method. The resulting cluster number is 3, with the number of PCA components being 7, which corresponds to 92.8% of the original data variance.

**Figure 7 sensors-22-09842-f007:**
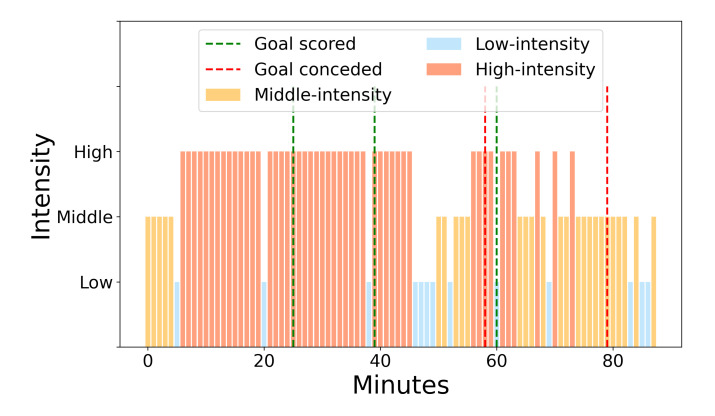
Intensity clusters across a 90-min period with the score-change timestamps for a single player in a particular game. High intensity is represented with brown–red, middle with yellow–orange, and low intensity with blue. Vertical discontinued red lines are goals conceded, and green ones are goals scored.

**Figure 8 sensors-22-09842-f008:**
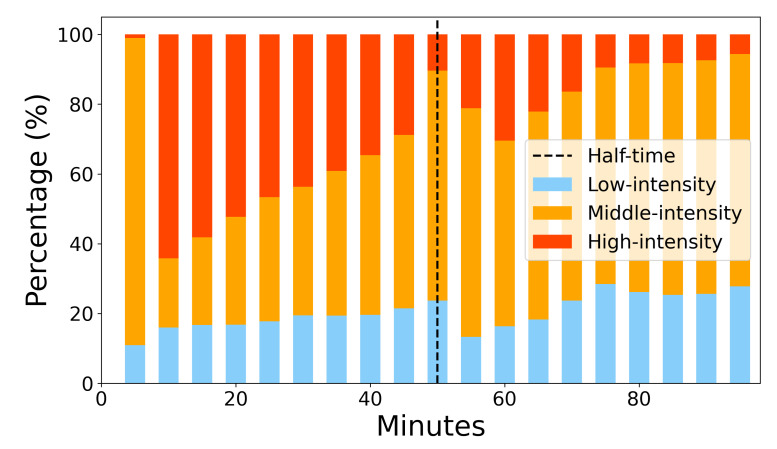
High-, middle-, and low-intensity cluster distribution across a period of a game in 5 min intervals.

**Table 2 sensors-22-09842-t002:** The most important features (above 0.3) for every PCA component.

Feature	PC1	PC2	PC3	PC4	PC5	PC6	PC7
Distance (m)	✓						
MPE count						✓	
Anaerobic energy (J/kg)	✓						
Average metabolic power (W/kg)	✓					✓	
Average MPE time (s)						✓	
Average MPE recovery time (s)						✓	
Walk energy (J/kg)	✓					✓	
Running energy (J/kg)	✓						
General energy (J/kg)	✓						
Total event count				✓			
Total energy spent (J/kg)				✓	✓		
MPE energy (3 min)		✓			✓	✓	
MPE energy (5 min)					✓		
Average recovery time (s) (3 min)					✓		
Average recovery time (s) (5 min)					✓		✓
Total recovery time (s)				✓			
MPE energy spent (J/kg)							✓
Event count		✓	✓				
Average recovery time (s)		✓	✓				

**Table 3 sensors-22-09842-t003:** Clustering group results. Darker background color represents higher intensity. The low-intensity group shows no MPE activity in the observed minute, but this is likely a result of very high expenditure in the 5 min period that preceded it. The middle-intensity group exhibits higher expenditure according to MPE features but shows lower values of GM-5MIN in 5 min prior. The high-intensity group produces the highest expenditure in all types of features.

Parameter Name	Low Group	Middle Group	High Group
MPE features (1 min)			
	μ σ	μ σ	μ σ
Energy (J)	0 0	180 150	230 280
Event count	0 0	1.8 0.9	2 1
Average recovery time (s)	60 0	20 7	18 8
MPE features (3 min before)			
	μ σ	μ σ	μ σ
Energy (J/kg) 3 min	400 350	400 330	700 350
MPE count (3 min)	3.8 2.2	4.0 2.2	6 2
Recovery time (s) (3 min)	155 16	154 15	140 16
GM-5MIN features (5 min before)			
	μ σ	μ σ	μ σ
Energy (J/kg)	2900 1000	1300 1000	2800 500
MPE count	6 3.5	4 3.5	9 2.5
Anaerobic energy (J/kg)	750 400	500 400	1000 150
Avg. MPE recovery time (s)	60 80	50 60	23 8
Running energy (J/kg)	1400 800	1000 800	2250 450

**Table 4 sensors-22-09842-t004:** Comparison of clustering algorithm and GM-GAME data for describing game load (63 games) based on CV. Bolded values mark higher explanatory power for a particular cluster.

Game Id	Clustering with K-Means			GM-GAME Data		
	**High**	**Middle**	**Low**	**High**	**Middle**	**Low**
1	0.3572	0.4042	0.2538	0.0743	0.5845	0.4497
2	0.2456	0.5178	0.2507	0.0881	0.5116	0.4089
3	0.3300	0.5092	0.1737	0.0511	0.5717	0.4257
⋮	⋮	⋮	⋮	⋮	⋮	⋮
79	0.4338	0.3946	0.1863	0.0679	0.6124	0.4352
80	0.2611	0.3997	0.3487	0.0695	0.5011	0.3963
μ±σ	0.31±0.14	0.51±0.16	0.20±0.05	0.07±0.01	0.57±0.1	0.42±0.08
CV	**45.91% **	**30.66%**	**24.41%**	21.32%	16.82%	19.12%

## Data Availability

Not applicable.

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
