# Peer review of "Extended Energy-Expenditure Model in Soccer: Evaluating Player Performance in the Context of the Game"

_sensors, 2022, doi:10.3390/s22249842_

Round 1

Reviewer 1 Report

General comment.

This document provide some interesting idea and point of view in potential new way to interpret data collected with gps system during soccer training and game. Nevertheless, many methodological points must be clarified and justify. In addition, an important lake of consistency over the document with terminology: game vs matches, intensity concept …, section vs chapter.

From my point of view, I highly suggested you to split this one in 2 separated studies.  A first one about your approach to clarify the aim and describing results carefully and helping potential users (coach, physical trainer, …) to understand the concept. Clustering, and many methods used in this study are not well presented and biggest part of reader will not understand concepts.

Second point do comparing clusters and interpret results between clusters inside games or over the season can a second or multiple studies.

Your document referring too much to subsection. If you document is constructed logically, you do not need that. That’s reinforcing the point that document must be split.

Clarify also averaging methods because recent methods mainly used rolling average where you seems to use different methods.

Your introduction is supported by many studies using Metrics from distance and speed but your concept is based on MP approach. I do not have any problem with that but you must be in line between section and potentially provide more detail in methodology.

English must be carefully checked, especially conjugation.

This document seems to be written by multiple authors with different style as a master thesis document and provide too big document with too many idea impossible to explain in single study.

There is not information about ethical agreement as required by the journal.

Line 17 – 22. All these claim must be support by references. There is a lot for soccer.

Line 26. This is not always true because it heavily depends on the context and the play styles of each team [3]. This sentence could be written based on statistics from references in paragraph inserted over. There is large CI providing contrasting results.

Line 27. Are you sure that fixtures is the best word in this context. Period may be more appropriated.

Line 28-29 Add a reference in this one a quote the authors to explain their findings.

Line 41.  Several works describes  by Several studies described

Line 42-43. I understand, the idea behind the sentence, but deeper is not appropriated. Almost every paper aim to go deeper, but not in your differentiation.  Some studies also evaluation the decline or organization of effort.

Dalen T, Lorås H, Hjelde GH, Kjøsnes TN, Wisløff U. Accelerations - a new approach to quantify physical performance decline in male elite soccer? Eur J Sport Sci. 2019 Sep;19(8):1015-1023. doi: 10.1080/17461391.2019.1566403. Epub 2019 Jan 11. PMID: 30632940.

Line 49 - this sentence must be refreshed - Not every game is the same, thus it should be treated like that

Line 59. Could instead of can. You are at hypothesis state.

Line 64. There is not chapter in a study, section may be more appropriate and in line with section 2 over.

Line 66 -72. Article do not need a summary in the introduction, discussion and conclusion are made for that.

In the introduction, some information are missing to get better understanding. HSR what is sprint Threshold. HIA, how they are defined?  And same for some others.  This part is interesting but not very fluid, maybe could be managed to get something easier to read.  Previous studies must be inserted using the past instead of present and introduction of your idea must be more objective.  Soccer analysis based on GPS data is the biggest topic in number of articles. You aim to add an interesting new way to read and interpret data.

Line 74-78. This section is not required.  You can insert the global design here and let each subsection detail the rest. You can describe this. The resulting dataset consisted of 80 matches that included preseason friendlies More accurately for example in the design.  This point is not information about participants.

Line 81  m instead of cm.

Line 82. Maybe, explaining that you exclude GK and include substitute maybe clearer. Did you include substitute without any restriction of duration played?

Any detail about ethical standard?

Line 89-92. Not required. GO direct in subsection.

Line 96. Reference from the company is required. (Exelio, Udine, Italy)  Devices instead of device.

Depending the threshold settled you can have more than 100 metrics…  Here I suggested to refresh with something like this.  

The matches’ data were collected using GPS systems GPexe pro2 device with sample rate of  18.8 Hz (Exelio Srl, Udine, Italy). After sessions, data collected was download from device and upload to the manufacturer platform using gpexe bridge application (Version…). The online web app gpexe computed all metrics using following settings (speed, HR, acc, decc…..)

Describing metrics, including MPE is mandatory because there are very specific and related to di Prampero works and not universal with overall of manufacturer.

Line 98 to 105 related to processing and interpretation. It must be place after data collection. This part is inserted at line 145 and very redounding.

Line 166.-207

There is no justification of metrics used or excluded. No information about threshold.

The entire introduction is based on examples using speed and distance approach and your article use mainly  MP approach.

Line 220. Parameters instead of features.

Line 229 -231. This sentence can be refresh and merged with next one to clarify and shorter.

Line 264. Use games instead of matches over the document.

Line 266 periods

Processing section if very long and redounding.  This entire section must be reformulated and clarify. No subjective description, opinion or other must be insert. Only methodology.

Line 319 – 324. Figure description must be place in figure information and not in the text.

Line 358- You mentioned previously introduction that data could be highly related to the context, opponent… and others.

Line 362. Interesting point but to long period induce lake of sensitivity and difficult to use and too short may be too much context related.

Line 397. Study instead of paper.

For all the discussion, previous concerned must be checked before.

Author Response

This document provide some interesting idea and point of view in potential new way to interpret data collected with gps system during soccer training and game. Nevertheless, many methodological points must be clarified and justify. In addition, an important lake of consistency over the document with terminology: game vs matches, intensity concept …, section vs chapter.

From my point of view, I highly suggested you to split this one in 2 separated studies.  A first one about your approach to clarify the aim and describing results carefully and helping potential users (coach, physical trainer, …) to understand the concept. Clustering, and many methods used in this study are not well presented and biggest part of reader will not understand concepts.

Second point do comparing clusters and interpret results between clusters inside games or over the season can a second or multiple studies.

Your document referring too much to subsection. If you document is constructed logically, you do not need that. That’s reinforcing the point that document must be split.

RESPONSE: We thank the Reviewer for expressing their concerns about splitting the paper into two or more studies. However, this is a new approach to data processing, and we believe that the inability to show the benefits and results of such processing would seem very incomplete - which would happen if we separated this into a new paper. Processing and clustering sections are vital for understanding the power of such an approach. However, without the practical application using the MFit index, they are not useful. We wanted to show that our framework for data processing creates new possibilities and options for better information extraction. Therefore, we have considered the Reviewer’s concerns in terms of changing the structure for the Methodology part where we tried to make it clearer and easier to read. This enables easier navigation through the paper and an easier understanding of the process.

Clarify also averaging methods because recent methods mainly used rolling average where you seems to use different methods.

RESPONSE: We thank the Reviewer for the comment. Sections 2.6.2 MFit Analysis and 3.2 MFit analysis: tracking players’ fitness status provides information about the averaging methods for the MFit index. We have also used rolling mean (average) but the terminology was not consistent. Therefore, section 2.6.2 was adjusted to make it clearer.

Your introduction is supported by many studies using Metrics from distance and speed but your concept is based on MP approach. I do not have any problem with that but you must be in line between section and potentially provide more detail in methodology.

RESPONSE: We appreciate the suggestion. In the extended section “Procedures and Variables”, we emphasized that our approach does not require setting any specific thresholds. Instead, we used energy-based metrics and total distance, and these features were described in more detail in the newly added part of the section. Moreover, the section Introduction was adapted to include the threshold concern in the text.

English must be carefully checked, especially conjugation.

This document seems to be written by multiple authors with different style as a master thesis document and provide too big document with too many idea impossible to explain in single study.

RESPONSE: We thank the Reviewer for the comment. We solved the problem concerning the inconsistency and style differences in the following way. First, one of the authors read the entire document and adjusted the text. Other authors then checked the document again and made corrections where necessary.

There is not information about ethical agreement as required by the journal.

RESPONSE: We thank the Reviewer for pointing this out. The ethical agreement section was added, stating that in our case it is not applicable because written consent from the club’s representative was provided. Moreover, the soccer club collected the data and shared it with the researchers involved in this study through a Non-Disclosure Agreement. Actually, the owner of the data is the elite soccer club that wants to remain anonymous. The club has the right to choose which information, results, and data can be made publicly available and has granted access to these data to the authors of this paper only for research purposes.

Line 17 – 22. All these claim must be support by references. There is a lot for soccer.

Line 26. This is not always true because it heavily depends on the context and the play styles of each team [3]. This sentence could be written based on statistics from references in paragraph inserted over. There is large CI providing contrasting results.

Line 27. Are you sure that fixtures is the best word in this context. Period may be more appropriated.

Line 28-29 Add a reference in this one a quote the authors to explain their findings.

Line 41.  Several works describes  by Several studies described

Line 42-43. I understand, the idea behind the sentence, but deeper is not appropriated. Almost every paper aim to go deeper, but not in your differentiation.  Some studies also evaluation the decline or organization of effort.

Dalen T, Lorås H, Hjelde GH, Kjøsnes TN, Wisløff U. Accelerations - a new approach to quantify physical performance decline in male elite soccer? Eur J Sport Sci. 2019 Sep;19(8):1015-1023. doi: 10.1080/17461391.2019.1566403. Epub 2019 Jan 11. PMID: 30632940.

Line 49 - this sentence must be refreshed - Not every game is the same, thus it should be treated like that

Line 59. Could instead of can. You are at hypothesis state.

Line 64. There is not chapter in a study, section may be more appropriate and in line with section 2 over.

Line 66 -72. Article do not need a summary in the introduction, discussion and conclusion are made for that.

In the introduction, some information are missing to get better understanding. HSR what is sprint Threshold. HIA, how they are defined?  And same for some others.  This part is interesting but not very fluid, maybe could be managed to get something easier to read.  Previous studies must be inserted using the past instead of present and introduction of your idea must be more objective.  Soccer analysis based on GPS data is the biggest topic in number of articles. You aim to add an interesting new way to read and interpret data.

Line 74-78. This section is not required.  You can insert the global design here and let each subsection detail the rest. You can describe this. The resulting dataset consisted of 80 matches that included preseason friendlies More accurately for example in the design.  This point is not information about participants.

The entire introduction is based on examples using speed and distance approach and your article use mainly  MP approach.

RESPONSE: We thank the Reviewer for the detailed comments. All the named issues, together with a mention of threshold-related problems were corrected and included in the new version of the Introduction section.

Line 81  m instead of cm.

Line 82. Maybe, explaining that you exclude GK and include substitute maybe clearer. Did you include substitute without any restriction of duration played?

Any detail about ethical standard?

Line 89-92. Not required. GO direct in subsection.

Line 96. Reference from the company is required. (Exelio, Udine, Italy)  Devices instead of device.

Depending the threshold settled you can have more than 100 metrics…  Here I suggested to refresh with something like this.

The matches’ data were collected using GPS systems GPexe pro2 device with sample rate of  18.8 Hz (Exelio Srl, Udine, Italy). After sessions, data collected was download from device and upload to the manufacturer platform using gpexe bridge application (Version…). The online web app gpexe computed all metrics using following settings (speed, HR, acc, decc…..)

Describing metrics, including MPE is mandatory because there are very specific and related to di Prampero works and not universal with overall of manufacturer.

Line 98 to 105 related to processing and interpretation. It must be place after data collection. This part is inserted at line 145 and very redounding.

RESPONSE: Materials and Methods section structure was significantly changed to make it more readable. This included introducing or modifying the following subsections: (1) Study Design, (2) Subjects, (3) Data Acquisition, (4) Procedures and Variables, (5) Data Preprocessing, and (6) Data Analysis. Study Design provides general information about the collection process, playing positions that were analyzed, and the two data sources that were used for creating the dataset. The Subjects section provides detail about the players and which positions were included in the analysis. The Data Acquisition section describes the type of devices, platform, and duration of the acquisition period. The procedures section was extended with Procedures and Variables to explain the information concerning the GPS metrics that were used in the analysis and how they were calculated. 

Line 166.-207

There is no justification of metrics used or excluded. No information about threshold.

Line 220. Parameters instead of features.

Line 229 -231. This sentence can be refresh and merged with next one to clarify and shorter.

Line 264. Use games instead of matches over the document.

Line 266 periods

Processing section if very long and redounding.  This entire section must be reformulated and clarify. No subjective description, opinion or other must be insert. Only methodology.

RESPONSE: We thank the Reviewer for pointing this out. The justification of metrics was changed and moved to a new section Procedures and Variables. All occurrences of the word “match” were replaced with the word “game”. The Data Processing section underwent major changes, heavily reducing the text and replacing portions of it with data source labels in Table 1 (GM-5MIN-PRIOR, MPE-PRIOR, MPE-CURRENT). The whole methodology part (section 2) was rewritten with the aim to make the processing section more fluid and easier to understand. 

Line 319 – 324. Figure description must be place in figure information and not in the text.

Line 397. Study instead of paper.

RESPONSE: We thank the Reviewer for the comment. The description of the Figure was moved to Figure 7. description. Terminology throughout the paper was changed to use “study” instead of “paper” wherever suitable.

Line 358- You mentioned previously introduction that data could be highly related to the context, opponent… and others.

RESPONSE: We thank the Reviewer for this comment. Now that we have added better clarification of the metrics used and made the Introduction and Processing sections easier to read we believe that it’s evident that our energy-based approach is less sensitive to context than the threshold-based metrics. We have added a statement to make this clear in section 3.2 MFit analysis: tracking players’ fitness status. 

Line 362. Interesting point but to long period induce lake of sensitivity and difficult to use and too short may be too much context related.

RESPONSE: We thank the Reviewer for pointing this. Based on the suggestion, we added a couple of sentences that better describe the problem of the duration length when looking at the maximum number of high-intensity consecutive repetitions.

For all the discussion, previous concerned must be checked before.

RESPONSE: We thank the Reviewer for the comment. The Discussion section was adjusted and corrected to be in line with the main concerns of the study.

Reviewer 2 Report

In this manuscript, the authors present a framework for modeling energy expenditure based on GPS wearable device data, introducing the MFit index to analyze the state of athletes in low-, medium- and high-intensity competitions. This is an interesting study, but there are certain limitations. Therefore, I would like to recommend that this manuscript be accepted with major revisions.

1. What are the advantages of using MFit index with a single value in a manuscript over the widely used GM- MATCH parameter? What are the advantages in terms of accuracy or breadth of application?

2. The energy consumption model based on MFit index has some limitations, such as the team's possession style, the quality of the opponent, and the substitution of players. How can these factors be taken into account in the clustering analysis and MFit analysis and how can they be optimized? (not excluding these factors)

3. A feature of the MFit index-based energy expenditure model is the minute-by-minute match analysis, which can be performed individually or as a team. What about the specific part about team match analysis?

4. There are some misspellings in the manuscript, such as "conceded" in Figure 7.

5. A comparison between the clustering algorithm and GM-MATCH in the manuscript mentions that " On the other hand, GM-MATCH puts focus on middle- and low-, with a very little influence of high-intensity. According to this, soccer players are not giving their maximal effort very often in competitive matches which is hard to believe.". Judging an athlete's performance based solely on the change in a numerical value without taking into account the athlete's physical condition, the quality of the opponent, or the environment is clearly lacking some consideration and unconvincing.

Author Response

In this manuscript, the authors present a framework for modeling energy expenditure based on GPS wearable device data, introducing the MFit index to analyze the state of athletes in low-, medium- and high-intensity competitions. This is an interesting study, but there are certain limitations. Therefore, I would like to recommend that this manuscript be accepted with major revisions.

What are the advantages of using MFit index with a single value in a manuscript over the widely used GM- MATCH parameter? What are the advantages in terms of accuracy or breadth of application?

RESPONSE: GM-MATCH parameters are used by the coaches for providing feedback regarding the volume and intensity of a training session or a game. However, there exists no consensus concerning the choice of parameters that need to be tracked to represent the game appropriately. Usually, only a handful of parameters are being tracked (such as HSR distance, sprint distance, total distance, acc/dec count, etc.) and the conclusions are extracted by considering them independently. The main advantage of the MFit index is that you don’t need to go through the feature engineering phase and pick the right parameters that you want to track, because a clustering method behind the MFit index already incorporates many parameters before making a decision. The result is a single value that is much easier to use. We added this statement to section 3.2 Mfit analysis: tracking players’ fitness status, together with a reference to a STATSports article that explains the use of analytical methods, on GM-MATCH data, to describe macrocycles.

The energy consumption model based on MFit index has some limitations, such as the team's possession style, the quality of the opponent, and the substitution of players. How can these factors be taken into account in the clustering analysis and MFit analysis and how can they be optimized? (not excluding these factors)

RESPONSE: We thank the Reviewer for a valid and interesting argument. The quality of the opponent surely affects the team’s possession style and produces different game demands. However, the data that we had on disposition did not include event data and the players’ positions at each moment in the match. Therefore, any information about the team’s specific playing or possession style was unavailable. The influence of the quality of the opponent requires additional research and a careful definition of the quality parameter. The best approach would be to use coefficients from the betting sites before the game, but these values are very hard or impossible to extract (for the observed team) in a reversed time frame. The remaining limitation regarding the substitution of players needs a bigger dataset with different teams so we could compare the MFit index from different players according to their playing position and see how they differ from a standard. All in all, the Mfit index could surely be improved but it requires a bigger dataset and additional exploration. We believe that these factors are out-of-scope of this paper and have therefore added these concerns in a new paragraph of the Discussion section.

A feature of the MFit index-based energy expenditure model is the minute-by-minute match analysis, which can be performed individually or as a team. What about the specific part about team match analysis?

RESPONSE: We thank the Reviewer for pointing this out. In the presented paper we explain a new framework for game data processing and analysis with the usage of the MFit index. We mention team analysis as a possibility and provide an idea of usage in the added paragraph in section 3.1.2 In-depth game visualization. We do not want to go deeper into team analysis because we want the focus of the paper to be on the processing methods and their application. However, the framework presented in this paper will provide a very good basis for further analysis of the team performance and inspection of the most suitable methods for aggregation, while keeping the most important information.

There are some misspellings in the manuscript, such as "conceded" in Figure 7.

RESPONSE: The manuscript was revised to correct all the misspellings. 

A comparison between the clustering algorithm and GM-MATCH in the manuscript mentions that " On the other hand, GM-MATCH puts focus on middle- and low-, with a very little influence of high-intensity. According to this, soccer players are not giving their maximal effort very often in competitive matches which is hard to believe.". Judging an athlete's performance based solely on the change in a numerical value without taking into account the athlete's physical condition, the quality of the opponent, or the environment is clearly lacking some consideration and unconvincing.

RESPONSE: We thank the Reviewer for raising the question on the comparison analysis. The goal of such a statement was to put emphasis on the problem of no consensus regarding GM-GAME (renamed from GM-MATCH) metrics which need to be tracked for controlling players' performances during training sessions and games. We wanted to show how difficult it is to use a single GM-GAME value to represent the game demands. Also, the usage of the MFit index makes it much easier for a practitioner to assess a player’s performance by inspecting only a single value. Additional grouping according to home or away, and higher or lower quality of the opposition can be done to extend the analysis, but the aim of this paragraph was to show the expressive power of the clustering and the MFit approach presented in this paper. To make this more clear, we elaborated on the goals of the analysis in section 3.1.3. Evaluation through game load.

Round 2

Reviewer 1 Report

Dear,

Thank you for your updates and comments. Your effort have been appreciated.

Document edition in not in line with the journal requirement. Document must be prepared following the journal guideline and updates must be highlighted to help reviewer to check these ones. Your document is very hard to follow and review. I your very long document, review is very difficult, and insertion are very difficult to follow.

I remain on same idea that this paper is too heavy with too much idea presenting an interesting method with too mush proposal which seems to be far away from real world. Potential utilization and real utilization from coaches and physical trainers is very different concept.

I believe that this methodology can provide interesting way to interpret intensities, or player profiles but other potential use or interest seem to be too much overground.  

Specific comment

Line 29. Strength  ???  quality or level maybe?

Line 32-33 remove fixture.

Line 50. In most cases the user can determine thresholds. Manufacturer can apply different methodology but threshold are set by user. Where are threshold for number of acceleration or deceleration???? Acceleration determination can have thresholds not the number of accelerations.

Line 76. Information about the gps system Pro 2 apparently and manufacturer (Exelio, Udine, Italy)

Line 85. Weight with .1 accuracy can be better than unit.

Line 112  total time played (unit)

Line 116  Average metabolic power was calculated by multiplying speed and energy cost,??? Are you sure about that????

Line 125. This sentence is inappropriate. MP decrease can be a consequence of game outcome and or physiological limitation. This is not a consequence of oxygen dept.

Line 138 -140. Any consensus for MPE calculation and threshold?

Line 172-174. Any reference or relevance to support this sentence and 5 min period choice?

Line 240. Why was 200 meters selected and why eliminate those periods?

Author Response

POINT 1: Document edition in not in line with the journal requirement. Document must be prepared following the journal guideline and updates must be highlighted to help reviewer to check these ones. Your document is very hard to follow and review. I your very long document, review is very difficult, and insertion are very difficult to follow.

I remain on same idea that this paper is too heavy with too much idea presenting an interesting method with too mush proposal which seems to be far away from real world. Potential utilization and real utilization from coaches and physical trainers is very different concept.

I believe that this methodology can provide interesting way to interpret intensities, or player profiles but other potential use or interest seem to be too much overground.  

RESPONSE: We thank the Reviewer for the comment. We strongly believe that the applicability of the approach has to be presented in order to prove its value. However, we understand that this particular part might be interesting to a specific reader, and confusing to ones not coming from coaching and conditioning grounds. Therefore, we have opted for a solution of moving all the MFit-related content to the Supplementary Material. We mention it in Section 2.7 for the ones that could be more interested in how the framework can be used for tracking players’ fitness.

POINT 2: Specific comment

Line 29. Strength  ???  quality or level maybe?

Line 32-33 remove fixture.

Line 50. In most cases the user can determine thresholds. Manufacturer can apply different methodology but threshold are set by user. Where are threshold for number of acceleration or deceleration???? Acceleration determination can have thresholds not the number of accelerations.

Line 76. Information about the gps system Pro 2 apparently and manufacturer (Exelio, Udine, Italy)

Line 112  total time played (unit)

RESPONSE: We thank the Reviewer for the specific comments. All the named issues were addressed and corrected in the text.

POINT 3: Line 85. Weight with .1 accuracy can be better than unit.

RESPONSE: We thank the Reviewer for the comment. Unfortunately, we don’t have the information regarding the weight that is more precise than the presented values.

POINT 4: Line 116  Average metabolic power was calculated by multiplying speed and energy cost,??? Are you sure about that????

Line 125. This sentence is inappropriate. MP decrease can be a consequence of game outcome and or physiological limitation. This is not a consequence of oxygen dept.

RESPONSE: We appreciate the suggestions. Both sentences are extracted from the internal GPexe documents that are explaining metrics for athletic performance monitoring. The document is only available to GPexe users. We have added a footnote to point this out.

POINT 5: Line 138 -140. Any consensus for MPE calculation and threshold?

RESPONSE: We thank the Reviewer for expressing their concerns. Unlike other GPS metrics, MPE based approach was presented by di Prampero et al. which gives a scientific verification of the approach and sets the standard for the others. A particular threshold does not exist, because it is dependent on the athlete’s maximum VO2. This can be directly or indirectly assessed as we have noted in the text.

POINT 6: Line 172-174. Any reference or relevance to support this sentence and 5 min period choice?

RESPONSE: We thank the Reviewer for pointing this out. Our attempts in extracting the data in a period of fewer than 5 minutes failed, most probably due to the internal limitation of the GPS provider and its server infrastructure. We understand that GPS collects a lot of data and the processing time of small intervals becomes very difficult.

POINT 7: Line 240. Why was 200 meters selected and why eliminate those periods?

RESPONSE: We thank the Reviewer for raising this issue. We chose this threshold experimentally to eliminate short periods just before the start of the game, as well as the ones at halftime. These values should not be used to create the clustering model - because their periods are not a part of the game. Once the model has been set (trained) then there is no problem in classifying these segments, and the result would most probably always be low-intensity.

Reviewer 2 Report

The revised manuscript is acceptable.

Author Response

We would like to thank the reviewer for his/her useful comments.